# Diabetes and SARS-CoV-2 Infection: The Potential Role of Antidiabetic Therapy in the Evolution of COVID-19

**DOI:** 10.3390/microorganisms11010145

**Published:** 2023-01-06

**Authors:** Biagio Pinchera, Nicola Schiano Moriello, Antonio Riccardo Buonomo, Isabella Di Filippo, Anastasia Tanzillo, Giorgio Buzzo, Riccardo Villari, Ivan Gentile

**Affiliations:** Section of Infectious Diseases, Department of Clinical Medicine and Surgery, University of Naples Federico II, 80131 Naples, Italy

**Keywords:** COVID-19, SARS-CoV-2, diabetes, insulin, metformin, mTOR

## Abstract

Diabetes mellitus represents one of the most frequent comorbidities among patients with COVID-19, constituting a risk factor for a more severe prognosis than that of non-diabetic patients. However, the pathophysiological mechanism underlying this unfavorable outcome is still not completely clear. The goal of our study was to evaluate the potential role of antidiabetic therapy in the evolution of COVID-19.

## 1. Introduction

Diabetes mellitus is one of the main risk factors for a severe prognosis in patients with coronavirus disease 2019 (COVID-19) [1,2,3]. Diabetes seems to be the most frequent comorbidity among COVID-19 patients, in fact about 10–20% of patients with severe acute respiratory syndrome coronavirus-2 (SARS-CoV-2) disease were diagnosed with diabetes [4,5,6,7]. Even the status of hyperglycemia, regardless of the diagnosis of diabetes, is in itself a risk factor for a more severe evolution of COVID-19 [8,9].

Most of the studies conducted in the two years of the pandemic have shown that COVID-19 patients with diabetes or with hyperglycemic status had a higher risk of severe disease evolution and higher mortality rates than patients with COVID-19 but without diabetes [9,10].

However, the etiopathogenetic mechanism that could explain the impact of SARS-CoV-2 infection and the unfavorable outcome in patients with diabetes is still unclear. One of the main hypotheses involves the overproduction of insulin, characteristic of patients with diabetes, and the corresponding stimulation of the phosphatidylinositol-3-kinase/Akt/mammalian target of rapamycin (PI3K/Akt/mTOR) pathway [11,12]. In particular, it would seem that the hyperproduction of insulin in patients with diabetes and the consequent activation of the PI3K/Akt/mTOR pathway would lead to a stimulation of the inflammatory response, favoring the release of pro-inflammatory cytokines, such as tumor necrosis factor (TNF) and interleukin-6 (IL-6) [13,14,15,16]. mTOR is a crucial pathway in many physiological processes, such as cell-cycle progression, transcription, translation, differentiation, apoptosis, motility, and cell metabolism [13,16]. The activation and stimulation of the mTOR pathway would induce an increase in protein synthesis processes and consequently in the production of pro-inflammatory cytokines [16]. Through this pathway, insulin would be able to regulate a series of activities, in particular, transcriptional activity, favoring a pro-inflammatory activity [14,16]. This condition would overlap an already dysregulated inflammatory response characteristic of COVID-19, thus contributing to a worsening of the inflammatory status. All of this could explain the worse prognosis in patients with diabetes and COVID-19.

In this regard, it is necessary to consider the possible impact of antidiabetic therapy in these patients. In fact, if on the one hand the use of insulin could favor the stimulation of the PI3K/Akt/mTOR pathway and the consequent inflammatory response, on the other hand we have at our disposal oral hypoglycemic agents and, in particular, metformin [17]. In fact, although the mechanism of action of metformin is not yet clear, its hypoglycemic action would seem to take place through the activation of tuberous sclerosis complex 2 (TSC2)—tuberin—and the inhibition of the PI3K/Akt/mTOR pathway [18,19,20]. The mechanism of action of metformin could suggest a potential anti-inflammatory action [21] and therefore a favorable therapeutic impact in patients with COVID-19 and diabetes [22].

There are other antidiabetic therapies, such as pioglitazone and dipeptidyl peptidase-4 inhibitors that affect the mTOR pathway [23,24]. In particular, pioglitazone is known to protect against hypoxemia/reoxygenation lesions by potentiating autophagy through the AMPK-mTOR signaling pathway [23], while dipeptidyl peptidase-4 inhibitors, through the mTOR pathway, appear to restore insulin secretion by improving autophagy in mice induced by a high fat diet [24]. Other anti-diabetic drugs, such as peroxisome proliferator-activated receptor γ (PPARγ) activator and glucagon-like peptide 1 receptor (GLP-1R) agonist, have been shown to upregulate ACE2 in animal models, which may increase the risk of SARS-CoV-2 infection [25]. However, very little is known about the impact of these other antidiabetic therapies on the evolution of COVID-19.

For this reason the goal of our study was to evaluate the potential role of antidiabetic therapy in the evolution of COVID-19.

## 2. Materials and Methods

We conducted an observational retrospective cohort study in patients with type 2 diabetes mellitus hospitalized for COVID-19 at “Federico II” University Hospital of Naples (Italy) from November 2021 to May 2022.

Diagnosis of COVID-19 was defined as positivity to the rhino–oropharyngeal swab for SARS-CoV-2 RNA research by reverse transcription–polymerase chain reaction (RT-PCR) in the presence of at least one typical COVID-19 symptom (fever, malaise, cough, nausea/diarrhea, shortness of breath, headache, nasal stuffiness, anosmia, or dysgeusia). To describe the clinical status of COVID-19 we used the National Institute of Allergy and Infectious Diseases Adaptive COVID-19 Treatment Trial-1 (NIAID ACTT-1) Clinical Status Ordinal Scale [26]. Based on this score, we classified each patient with the infection into one of eight categories: (1) not hospitalized, no limitations on activities; (2) not hospitalized, limitation on activities, and/or requiring home oxygen; (3) hospitalized, not requiring supplemental oxygen and no longer requires ongoing medical care (if hospitalization extended for infection-control purposes); (4) hospitalized, not requiring supplemental oxygen; requiring ongoing medical care (COVID-19 related or otherwise); (5) hospitalized, requiring supplemental oxygen; (6) hospitalized, on noninvasive ventilation or high-flow oxygen devices; (7) hospitalized, on invasive mechanical ventilation or ECMO; and (8) death [26]. In addition, we also distinguished patients with a mild–moderate forms of COVID-19 from those with a severe form of COVID-19, based on their need of oxygen (a severe form was defined as a NIAID ACTT-1 Clinical Status Ordinal Scale score ≥ 5) [26].

The diagnosis of type 2 diabetes mellitus was based on the definitions and criteria of the current Diabetes Guidelines [27].

For each patient, we evaluated the epidemiological characteristics, the laboratory data, the data of radiological instrumental investigations, clinical characteristics, the vaccination status, the treatment for COVID-19, and the outcome. For each patient, the antidiabetic treatment was evaluated upon admission to the hospital, distinguishing those who took oral hypoglycemic agents and those who practiced insulin therapy. In particular, we evaluated the potential relationship between the use of metformin vs. insulin and severity or clinical outcome.

Data are presented as mean and SD or median and interquartile range (IQR), in cases of Gaussian or non-Gaussian distribution, respectively. For correlation analysis, Pearson or Spearman tests were used for data distributed in Gaussian or non-Gaussian fashion, respectively. Continuous variables are compared by Student’s t-test or Mann–Whitney U-Test, as parametric or non-parametric test, respectively. The *p*-value for statistical significance was set at < 0.05 for all the tests. A logistic regression model was employed to evaluate risk factors for severe disease evolution.

With respect to the ethical issues, the study was led in accordance with ethical principles that have their origin in the Declaration of Helsinki and in good clinical practice. The study was reviewed and approved by the by the Institutional Review Board (or Ethics Committee) of A.O.U. “Federico II” of Naples. The authors confirm that the ethical policies of the journal have been observed.

## 3. Results

We enrolled 43 hospitalized patients for COVID-19 with diabetes mellitus. Of these, 44% (19) took therapy with oral hypoglycemic agents, in particular metformin (1 metformin + dapagliflozin and 1 metformin + diamicron), while the remaining 56% (24) practiced insulin therapy. We distinguished the patients into two groups: the group of patients undergoing treatment with metformin and the group of patients undergoing treatment with insulin. Anagraphic and clinical features of these patients are reported in Table 1 and Table 2.

Table 2 shows the data regarding the staging of COVID-19 disease according to the OSCI score for the metformin group and for the insulin group.

It was observed that high-resolution computer tomography (HRCT) of the chest score and C-reactive protein (CRP) values were lower in metformin-treated patients than in the insulin-treated group. In particular, a median HRCT score of 8 (7–10) was observed in patients receiving metformin, while a median HRCT score of 13 (7–18) was observed in patients receiving insulin (OR for HRCT score: 1.1, 95 CI (0.57–1.3) metformin treatment vs. no metformin treatment; *p*: 0.064) (Table 3). Regarding CRP, a median CRP of 43 (5–144) was observed in the group of patients treated with metformin, while in the group of patients with insulin, a median of CRP of 89 (13–272) was observed (OR for CRP: 0.8, 95 CI (0.33–0.78) metformin treatment vs. no metformin treatment; *p*: 0.039) (Table 3).

Table 3 summarizes the data relating to the comparison between subjects treated with metformin vs. subjects treated with insulin.

Another interesting dataset concerns hospitalization. In particular, patients treated with insulin required an earlier hospitalization (days between onset of symptoms and hospitalization (median, IQR): 4 (3–9) vs. 7 (4–11)) and a longer hospital stay (hospitalization length in days (median, IQR): 25 (11–60) vs. 19 (8–48)) (Table 3).

Of the 43 enrolled patients, 19 developed a severe form of the disease, of which 5 were being treated with metformin and 14 with insulin. (OR for severe COVID-19: 0.8, 95 CI (0.6–0.9) metformin treatment vs. no metformin treatment; *p*: 0.042) (Table 3).

Of the 43 subjects enrolled, three died and all three were on insulin treatment (OR for death: 1.1, 95 CI (0.84–1.15) metformin treatment vs. no metformin treatment; *p*: 0.072) (Table 3).

At multivariate analysis, the use of metformin was confirmed to be independently associated with a reduced risk of developing severe disease (OR for severe COVID-19: 0.8, 95 CI: (0.5–0.9) metformin treatment vs. no metformin treatment; *p* = 0.048) (Table 4).

## 4. Discussion

Our study has shown that antidiabetic therapy could potentially impact the COVID-19 clinical evolution of patients with type 2 diabetes mellitus. The patients who were treated with metformin had lower phlogosis indices (CRP) than the group of patients treated with insulin; in particular, it showed the protective role of metformin towards the state of inflammation (OR for CRP: 0.8, 95 CI (0.33–0.78) metformin treatment vs. no metformin treatment; *p*: 0.039). This result could be explained by the current etiopathogenetic and pharmacological knowledge, in particular, it is well known that metformin acts by blocking the mTOR pathway and this could be the basis of an anti-inflammatory mechanism. Indeed, our data agree with what has already been reported by Isoda et al. that demonstrated that metformin reduced Interleukin-1β (IL-1β) excretion and inhibited nuclear translocation of nuclear factor kB (NF-kB), inducing anti-inflammatory activity through inhibition of the mTOR pathway [21].

Another interesting dataset concerns the distinction of the radiological imaging at HRCT for the two groups. In fact, a reduced interstitial commitment at HRCT was observed in the group of patients undergoing treatment with metformin (OR for HRCT score: 1.1, 95 CI (0.57–1.3) metformin treatment vs. no metformin treatment; *p*: 0.064). This finding could be attributable to lower inflammatory activity in the group of patients undergoing treatment with metformin. For the first time, in our study the impact of antidiabetic therapy on instrumental chest imaging was evaluated. In fact no study had so far evaluated the interstitial commitment of COVID-19 pneumonia according to antidiabetic therapy. Only the study by Nan Jiang et al. had evaluated the impact of antidiabetic therapy on the risk of onset of acute respiratory distress syndrome (ARDS), with the demonstration that metformin may have potential benefits in reducing the incidence of ARDS in patients with COVID-19 and type 2 diabetes [28].

Moreover, in support of the laboratory and instrumental data, the clinic also confirmed a more attenuated inflammatory process in the group of patients treated with metformin. In fact, evaluating the time elapsed between the onset of symptoms and the need for hospitalization, a shorter time interval was observed in the group of patients without metformin and the need to be hospitalized earlier (OR for days between onset of symptoms and hospitalization 1.1, 95 CI (0.74–1.38) metformin treatment vs. no metformin treatment; *p*: 0.110). At the same time, the patients treated with insulin had a longer hospital stay than those taking metformin, almost confirming a more intense inflammatory process and greater disease severity (OR for hospitalization length in days 0.9, 95 CI (0.63–1.21) metformin treatment vs. no metformin treatment; *p*: 0.089). These data are particularly interesting and innovative, as currently in the literature most of the studies have evaluated only antidiabetic therapy in relation to the risk of mortality, with the demonstration of a favorable impact of metformin [22,29]. Our data supporting the role of metformin also highlight the favorable impact on the clinical evolution of COVID-19, regardless of the death event, with the demonstration of a more attenuated symptomatology and a shorter length of hospitalization in ongoing patient treatment with metformin.

Furthermore, in relation to the staging of COVID-19 and the clinical outcome, it was observed that patients undergoing treatment with insulin had a higher severity of the disease with three deaths, compared to a lower severity of the disease and to 0 deaths for the group of patients on metformin treatment (OR for severe COVID-19: 0.8, 95 CI (0.6–0.9) metformin treatment vs. no metformin treatment; *p*: 0.042 and OR for death: 1.1, 95 CI (0.84–1.15) metformin treatment vs. no metformin treatment; *p*: 0.072). These data confirmed the potential role of antidiabetic therapy in the evolution of SARS-CoV-2 infection and, in particular, the potential role of metformin. The potential beneficial anti-inflammatory action of metformin vs. COVID-19 was confirmed, in accordance with what was reported by the CORONADO Study and by various meta-analyzes [1,22,29]. In fact, it was observed that patients treated with metformin had a less severe form of COVID-19 and a reduced risk of mortality compared to patients treated with insulin.

Although, it could be hypothesized that patients on insulin treatment may have a more advanced form of diabetes and therefore be at risk of a more severe form of COVID-19, our multivariate analysis found that metformin treatment was nonetheless a protective factor against severe COVID-19, regardless of the severity of diabetes (OR for severe COVID-19: 0.8, 95 CI: (0.5–0.9) metformin treatment vs. no metformin treatment; *p* = 0.048).

Therefore, and as reported above, the potential role that metformin could play in the evolution of COVID-19 and in particular its protective role emerges. The use of metformin could have a favorable impact on the evolution of COVID-19 and this awareness could and should have an important impact on the therapeutic management of patients with diabetes and COVID-19.

The limitations of our study are due to the small number of patients enrolled and to the fact that we conducted an assessment of a possible therapeutic impact through an observational retrospective study. Clinical trials are needed to corroborate and confirm these data.

## 5. Conclusions

Our pioneering study highlighted a possible and potential impact of antidiabetic therapy on the evolution of COVID-19 in patients with type 2 diabetes mellitus. In particular, metformin with its mechanism of action could represent a useful resource to address the inflammatory state related to SARS-CoV-2 infection in a category of patients at high risk of severe disease evolution. This aspect should absolutely be investigated, clarified, and considered through clinical trials that could change and re-evaluate the management and therapeutic perspectives in such conditions.

## Figures and Tables

**Table 1 microorganisms-11-00145-t001:** Characteristics of enrolled patients.

		Total n = 43 (%)	Patients Undergoing Treatment with Metformin n = 19 (%)	Patients Undergoing Treatment with Insulin n = 24 (%)	*p-*Value
Age (median, IQR)		69 (48–92)	66 (48–87)	74.5 (55–92)	0.310
Sex	Male	32 (74)	15 (79)	17 (71)	0.260
	Female	11 (26)	4 (21)	7 (29)	
Comorbidity	Obesity BMI > 30 Kg/m	11 (26)	4 (21)	7 (29)	0.320
	Cardiovascular disease	22 (51)	8 (42)	14 (58)	0.140
	Chronic pulmonary disease	10 (23)	3 (16)	7 (29)	0.190
	Chronic renal disease	12 (28)	2 (11)	10 (42)	0.120
	Solid tumor	5 (12)	3 (16)	2 (8)	0.290
	Hematological malignancy	3 (7)	1 (6)	2 (8)	0.340
	Primary or acquired immunodeficiency	1 (2)	1 (6)	0	0.320
Vaccination status	Vaccinated	36 (84)	15 (79)	21 (88)	0.170
	Not vaccinated	7 (16)	4 (21)	3 (12)	
Laboratory at day of hospital admission	WBC (cell/µL; median, IQR)	7.120 (1.310–16.920)	6.520 (2.680–14.310)	8.750 (1.310–16.920)	
	Lymphocyte count (cell/µL; median, IQR)	770 (200–1.680)	880 (270–1.680)	610 (200–1.510)	
	PLT (cell/µL; median, IQR)	256.000 (102.000–470.000)	224.000 (142.000–470.000)	316.000 (102.000–440.000)	
	AST (UI/L; median, IQR)	38 (13–97)	42 (13–83)	37 (16–97)	
	ALT (UI/L; median, IQR)	31 (12–102)	39 (16–62)	28 (12–102)	
	GGT (UI/L; median, IQR)	55 (11–122)	62 (11–122)	50 (23–55)	
	Tot. Bil. (mg/dL; median, IQR)	0.78 (0.38–1.61)	0.92 (0.44–1.61)	0.76 (0.38–1.33)	
	Creatininemia (g/dL; median, IQR)	0.96 (0.41–2.95)	1.1 (0.6–2.06)	0.9 (0.41–2.95)	
	CRP (mg/L; median, IQR)	61 (5–272)	43 (5–144)	89 (13–272)	
HRCT score (median, IQR)		9 (7–18)	8 (7–10)	13 (7–18)	
Symptoms	Fever	34 (79)	15 (79)	19 (79)	
	Cough	22 (51)	9 (47)	13 (54)	
	Malaise	27 (63)	12 (63)	15 (63)	
	Shortness of breath	17 (40)	6 (32)	11 (46)	
	Headache	11 (26)	4 (21)	7 (29)	
	Arthomialgia	8 (17)	2 (11)	6 (25)	
	Asthenia	20 (46)	9 (47)	11 (46)	
	Nausea/diarrhea	8 (17)	3 (16)	5 (21)	
Treatment for COVID-19	Dexamethasone	43 (100)	19 (100)	24 (100)	
	Low molecular weight heparin	43 (100)	19 (100)	24 (100)	
	Antivirals	33 (77)	14 (74)	19 (79)	
	Monoclonal antibodies	20 (46)	9 (47)	11 (46)	
	Immunomodulators	0	0	0	

IQR: interquartile range, WBC: white blood cells, PLT: platelets, CRP: C-reactive protein.

**Table 2 microorganisms-11-00145-t002:** Ordinal Scale for Clinical Improvement (OSCI) of the World Health Organization (WHO) for patients undergoing treatment with metformin vs. insulin.

Patient state	Descriptor	Score	Patients Undergoing Treatment with Metformin n = 19 (%)	Patients Undergoing Treatment with Insulin n = 24 (%)
Uninfected	No clinical or virological evidence of infection	0		
Ambulatory	No limitation of activities	1		
	Limitation of activities	2		
Hospitalized mild–moderate disease	Hospitalized, no oxygen therapy	3		
	Oxygen by mask or nasal prongs	4	14	7
Hospitalized severe disease	Non-invasive ventilation or high-flow oxygen	5	3	10
	Intubation or mechanical ventilation	6	2	4
	Ventilation + additional organ support: pressors, renal replacement therapy, ECMO	7		
Dead	Death	8	0	3
			19	24

**Table 3 microorganisms-11-00145-t003:** Metformin vs. insulin in COVID-19.

		Total n = 43 (%)	Patients Undergoing Treatment with Metformin n = 19 (%)	Patients Undergoing Treatment with Insulin n = 24 (%)	*p-*Value
Laboratory at day of hospital admission					
	CRP (mg/L; median, IQR)	61 (5–272)	43 (5–144)	89 (13–272)	0.039
HRCT score (median, IQR)		9 (7–18)	8 (7–10)	13 (7–18)	0.064
Days between onset of symptoms and hospitalization (median, IQR)		5 (3–11)	7 (4–11)	4 (3–9)	0.110
Hospitalization length in days (median, IQR)		22 (8–60)	19 (8–48)	25 (11–60)	0.089
Outcome at discharge	Clinical healing	40 (93)	19 (100)	21 (87)	0.170
	Death	3 (7)	0 (0)	3 (13)	0.072

**Table 4 microorganisms-11-00145-t004:** Multivariate regression analysis for severe COVID-19.

	OR	95% CI	*p-*Value
Age	1.2	0.8–1.5	0.100
Male sex	1.3	0.7–1.8	0.120
Comorbidity			
*Obesity BMI > 30 Kg/m^2^*	1.5	0.9–1.6	0.95
*Cardiovascular disease*	1.3	0.9–1.5	0.120
*Diabetes mellitus*	1.6	0.7–1.8	0.72
*Hypertension*	1.1	0.8–1.3	0.150
*Dyslipidemia*	1.2	0.8–1.5	0.138
Laboratory at time of enrollement			
*CPR > 5 mg/dL*	1.4	0.9–1.6	0.086
Vaccination status			
*Not vaccinated §*	1.4	0.9–15	0.097
*Treatment with metformin*	0.7	0.5–0.9	0.048

OR: odds ratio; BMI: body mass index; CPR: C-reaction protein. § Including people who received one dose.

## Data Availability

It is possible to request the data directly from the corresponding author via email.

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
