# Peer review of "Diabetes and SARS-CoV-2 Infection: The Potential Role of Antidiabetic Therapy in the Evolution of COVID-19"

_microorganisms, 2023, doi:10.3390/microorganisms11010145_

Round 1

Reviewer 1 Report

This is an interesting and topical manuscript that aims to delineate the effects of diabetes therapy (and the resulting effects on inflammation) with Covid severity and clinical outcome. 

Overall, the paper is well written and the data analysis and conclusions are sound. The data clearly supports the observations that are made and the rationale for the effects of inflammation, based on the specific therapeutic, are reasonable. Inflammation biomarker (CRP) and imaging again supports the conclusions that are drawn.

Suggested Corrections:
There are a few typos (incorrect capitalization, stray punctuation) that need to be corrected.

And the top half of Table 3 is repetitive wrt Table 1. I would suggest moving the p values for Age, Sex, Comorbidity, and Vaccination Status to Table 1. Then Have Table 3 list the remaining data and p values

Author Response

Response to Reviewer 1

First, I thank the Reviewer 1 for the comments, reviews and suggestions. Your contribution and support have been fundamental and essential for the improvement of the manuscript.

I proceeded to carry out all the indicated and suggested revisions. Below are the specific revisions made, as also highlighted in the manuscript.

Response to Reviewer 1

Suggested Corrections:

There are a few typos (incorrect capitalization, stray punctuation) that need to be corrected.

I proceeded to carry out the required revisions, as reported in the manuscript.

And the top half of Table 3 is repetitive wrt Table 1. I would suggest moving the p values for Age, Sex, Comorbidity, and Vaccination Status to Table 1. Then Have Table 3 list the remaining data and p values.

I proceeded to carry out the required revisions, as reported in the manuscript.

I thank the Reviewer 1 for the fundamental contribution and essential support.

Thank you.

Reviewer 2 Report

Title : “Diabetes and SARS-CoV-2 infection: the potential role of anti-2 diabetic therapy in the evolution of COVID-19”

Main comments

   This manuscript studied and analyized the role of anti-diabetic drug Metformin in COVID-19 infection and its outcome. Author classified diabetic patients with various categories and evaluated the role of antidiabetic therapy. This study gives the insight of increased inflammation due to diabetics and COVID-19 morbidity. Author need to address few questions before the acceptance of the manuscript.

Questions

1.     In page 1, Line 18, Please Change the letter “I” in Corono Virus to lower case

2.     Could author explain more about PI3K / Akt / mTOR pathway and it’s recent citation?

3.     In this study, Metformin was the only anti-diabetics drug was used to compare the role of anti-diabetic therapy in the evolution of COVID-19. Is there any other anti-diabetic drugs studied for evolution of COVID-19 and how it changes with the current work? In addition, Page 1, Line 43-44, author has mentioned the mechanism of metformin was not yet clearly understood.

4.     Does author included the diabetics patient not infected with COVID-19 but take anti-diabetic drug metformin in this study and analysied data from them to compare with other tested groups?

5.     Did author any study or groups with the supplement of direct anti-inflammatory drugs in diabetic patient with and without COVID-19 infection and its outcome?

      6.  The discussion part was not clearly written. Please rewrite the major finding of this work in the discussion part. 

Author Response

Response to Reviewer 2

First, I thank the Reviewer 2 for the comments, reviews and suggestions. Your contribution and support have been fundamental and essential for the improvement of the manuscript.

I proceeded to carry out all the indicated and suggested revisions. Below are the specific revisions made, as also highlighted in the manuscript.

Response to Reviewer 2

Questions

  1. In page 1, Line 18, Please Change the letter “I” in Corono Virus to lower case

 I have done the requested review

  1. Could author explain more about PI3K / Akt / mTOR pathway and it’s recent citation?

I proceeded to carry out the requested revision, as reported in the manuscript and below.

“…mTOR is a crucial pathway in many physiological processes, such as cell cycle progression, transcription, translation, differentiation, apoptosis, motility and cell metabolism [13,16]. The activation and stimulation of the mTOR pathway would induce an increase in protein synthesis processes and consequently in the production of pro-inflammatory cytokines [16]. Through this pathway, insulin would be able to regulate a series of activities, in particular, transcriptional activity, favoring a pro-inflammatory activity [14,16]…”

  1. In this study, Metformin was the only anti-diabetics drug was used to compare the role of anti-diabetic therapy in the evolution of COVID-19. Is there any other anti-diabetic drugs studied for evolution of COVID-19 and how it changes with the current work? In addition, Page 1, Line 43-44, author has mentioned the mechanism of metformin was not yet clearly understood.

I proceeded to deepen the requested topic, as reported in the manuscript and below.

“…There are other antidiabetic therapies, such as Pioglitazone and dipeptidyl peptidase-4 inhibitors that affect the mTOR pathway. In particular, Pioglitazone is known to protect against hypoxemia/reoxygenation lesions by potentiating autophagy through the AMPK-mTOR signaling pathway [23], while dipeptidyl peptidase-4 inhibitors, through the mTOR pathway, appear to restore insulin secretion by improving autophagy in mice induced by a high fat diet [24]. Others anti-diabetic drugs, such as peroxisome proliferator-activated receptor γ (PPARγ) activator and glucagon-like peptide 1 receptor (GLP-1R) agonist, have been shown to upregulate ACE2 in animal models, which may increase the risk of SARS-CoV-2 infection [25]. However, very little is known about the impact of these other antidiabetic therapies on the evolution of COVID-19...”

Regarding the mechanism of metformin, in reality the mechanism of action is not yet clear. From the data currently available in the literature, it can be seen that its mechanism of action seems to be partly realized through the inhibition of key genes of cellular metabolism, among which, the most studied are those of the PI3K/Akt/mTOR pathway which through the activation of TSC2 (Tuberous Sclerosis Complex 2, tuberine) negatively regulate the activity of mTOR. This aspect is reported in the manuscript: “…In fact, although the mechanism of action of metformin is not yet clear, its hypoglycemic action would seem to take place through the activation of Tuberous Sclerosis Complex 2 – tuberine (TSC2) and the inhibition of the PI3K / Akt / mTOR pathway [18-20]. The mecha-nism of action of metformin could suggest a potential anti-inflammatory action [21]…”.

  1. Does author included the diabetics patient not infected with COVID-19 but take anti-diabetic drug metformin in this study and analysied data from them to compare with other tested groups?

Unfortunately we do not have a control group of patients not infected with SARS-CoV-2 and treated with antidiabetics. Since we are a Division of Infectious Diseases, only patients with SARS-CoV-2 infection accessed us.

  1. Did author any study or groups with the supplement of direct anti-inflammatory drugs in diabetic patient with and without COVID-19 infection and its outcome?

Unfortunately we do not have this data available.

  1. The discussion part was not clearly written. Please rewrite the major finding of this work in the discussion part.

I proceeded to expand and deepen as required.

I thank the Reviewer 2 for the fundamental contribution and essential support.

Thank you.
